# The Impact of Corporate Social Responsibility Implementation on Enterprises' Financial Performance—Evidence from Chinese Listed Companies

Xudong Li, Ali Esfahbodi * and Yufeng Zhang

Birmingham Business School, University of Birmingham, Birmingham B15 2TT, UK;
xxl308@alumni.bham.ac.uk (X.L.); y.zhang.6@bham.ac.uk (Y.Z.)
* Correspondence: a.esfahbodi@bham.ac.uk

**Abstract:** Along with the constant changes in the current business environment, more and more enterprises have recognised the importance of Corporate Social Responsibility (CSR). Considering that profit maximisation is the eternal pursuit of enterprises and that some studies have already linked the financial performance of enterprises and their implementation of social responsibility together, this study will try to further explore the impact of social responsibility initiatives on enterprises' financial performance within the context of emerging economy. Given that enterprises' sustainable development is closely related to their implementation of CSR, an improvement in their corresponding financial performance due to effective social responsibility practices can incentivise enterprises to take part in CSR initiatives aimed at enhancing the sustainable development of society and the environment. Through using the panel data from Chinese Listed Companies, this research finds that responsibility's implementation is positively related with enterprises' financial performance, and that relationship is non-linear. Additionally, as a critical regulatory institution, government fails to function as a mediator within the above-mentioned relationship based on the robust empirical test. At the same time, the fulfilment of CSR can not be achieved at the expense of profit maximisation. The non-linear relationship between CSR and enterprises' financial performance (CFP) demonstrated in this research suggests that the financial performance of a firm can be optimised when it moderately fulfils its social responsibility. This finding offers a potential optimal strategy for the sustainable development of the firm as well as society. Also, the role of government deserves further exploration and utilisation, considering its significant linkages with enterprises and social development.

**Keywords:** corporate social responsibility; financial performance; Chinese listed companies; stakeholder theory; sustainable development

## 1. Introduction

The attention towards Corporate Social Responsibility (CSR) can be traced back to the late 1990s, when almost all the constituencies of society, from public institutions to single individuals, realised the importance of CSR. Till the end of the 1990s, only about 10% of Fortune 500 firms had not regarded CSR as a critical component in their organisation's goals. In the meantime, many corporations have also realised the importance of CSR implementation and increased their CSR investment accordingly [1]. Some scholars claimed that enterprises should treat CSR just like a form of investment strategy, which highly emphasises the significance of CSR in the long-term vision of a firm [2]. Coelho, Jayantilal, and Ferreira [3] also support this assertion and point out that confirming the contribution of CSR implementation to enterprises' financial performance (CFP) is critical. It has been widely accepted that profit maximisation is the nature of enterprises [4]. Therefore, it is essential to answer whether the allocation of enterprises' resources to the field of 'good citizenship' can trigger the increase in business value because enterprises could have used those resources in other, more profitable ways [5]. In addition, as an emerging economy,

China has also paid much attention to CSR issues. For example, in 2006, the Shenzhen Stock Exchange issued 'Shenzhen Stock Exchange Social Responsibilities Introductions to Listed Companies' guidelines to encourage the disclosure of CSR issues in the annual report of Chinese Listed Companies [6]. Therefore, this study tries to further clarify the relationship between CSR-CFP by using the data of Chinese Listed Companies and probing whether good CSR practices can improve corporates' financial performance.

However, no consensus has been made regarding the relationship between CSR and CFP, and clarifying that relationship has become the main argument in recent CSR studies [7]. Some scholars have concluded that the difference in evaluation index, the choice of control variables, the omittance of key variables, and the sampling method can become the potential reasons for the conflicts of previous outcomes [8–12]. From the theoretical analysis and empirical deduction perspective, there is still a 'musk' covering the relationship between CSR-CFP. Therefore, this study tries to employ empirical research to further clarify that relationship within the contemporary business landscape.

Regarding the practical gap, most of the background of prior studies was located in developed countries while overlooking the conditions that happened in developing countries [13]. As highlighted by Partalidou et al. [14], the interaction between CSR and CFP is remains contentious and merits further exploration. The increasing number of studies that focus on CSR also call for a refreshed viewpoint based on the context of the emerging economies. As the Chinese economic level and public attention to Chinese enterprises' CSR implementation have reached a relatively high point, an investigation into the CSR-CFP relationship within China becomes pivotal. It can serve as a representative of developing countries' circumstances. The shift in a research context is helpful to expand the extant research on CSR and clarify the relationship between CSR and CFP through insights drawn from emerging economies. In addition, few studies have tested the non-linear relationship between CSR and CFP [15,16]. Given the condition of the current business environment, the author assumes that the nonlinearity could be detected in the CSR-CFP relationship. Therefore, this research tries to fill the practical gap through threshold regression analysis. Moreover, many studies have focused on the role of government intervention within the CSR-CFP relationship and empirically proved government intervention as a moderator in that relationship [17,18]. However, few studies have identified the mediating function of government intervention. This study tries to fill the gap by adopting government subsidy as a proxy for government intervention.

Addressing the theoretical gap, the stakeholder theory has been recognised as an essential foundation that can underpin the CSR-CFP relationship. It is plausible that many scholars have realised the efficacy of stakeholders' perspectives regarding CSR measurement [19]. Nevertheless, the majority of the previous literature adopted the questionnaire or third-party ratings to gauge the CSR performance of enterprises [13,20–23]. The previous measurements of CSR performance have shown limitations in terms of objectivity, as ensuring consistency across different viewers can be challenging. Therefore, the comparability and robustness of the outcomes may be undermined. Moreover, Eabrasu [24] claimed that using subjective criteria to evaluate CSR performance will lead to a biased view. In the meantime, Cho and Lee [25] also appealed to use actual expenditure on CSR performance to replace KLD or other third-party ratings. Hence, the subjectivity of CSR measurement becomes the crux of the theoretical gap, and this study will use objective data to overcome it.

This paper is expected to contribute to the existing literature within the research field of CSR from the following dimensions: Firstly, this research expands the horizon of previous CSR research. The extant studies regarding the CSR-CFP relationship are primarily confined to large corporations in developed countries [13,22,26]. Through changing the perspective from developed countries to emerging economies, this study enlarges the inclusiveness of current CSR issues based on the evidence from emerging economies. Secondly, according to Ip [27], the unique regulatory environment of the Chinese market provides a brand-new context for CSR study. Additionally, the focus of CSR issues has

been changed from pure ethical considerations to multiple dimensions (e.g., social impact, stakeholders' concerns, etc.). Starting from the stakeholder theory, this study can evaluate the effect of CSR issues on corporates' financial performance from various perspectives. In addition, by using objective data to measure enterprises' CSR performance from multiple stakeholders' views, the measurement subjectivity of previous studies can be undermined, which is expected to increase the accuracy and robustness of the outcomes [20,23,28]. Last but not least, one of the strategic meanings of CSR study is enhancing corporate performance [29]. After conducting the basic regression and threshold analyses, a desirable CSR level that can achieve the balance of social and financial interests is expected to be confirmed. Considering the limited resources of enterprises, this provides a good reference for enterprises to conduct sound business in the CSR framework, which is conducive to achieving mutual development for both the enterprises and the whole society.

According to the above discussion, the research questions for this paper are as follows: (1) Does the good practice of CSR trigger a positive influence on the enterprises' financial performance within the context of Chinese Listed Companies? (2) Is this influence linear one or a non-linear? (3) Does the government subsidy function as a mediator within the CSR-CFP relationship?

The remaining parts of this study are organised as follows: Section 2 reviews the relevant literature and posits the corresponding hypotheses. Section 3 presents the data and regression model. Section 4 analyses the empirical findings. Section 5 concludes the whole research. And Section 6 illustrates the potential limitations and future direction.

## 2. Theoretical Foundations

### 2.1. Corporate Social Responsibility

Until now, numerous scholars have had various debates regarding the definition of CSR. In 1953, Bowen took the lead to theorise the relationship between society and corporations in his seminal book, *Social Responsibilities of the Businessman*. He delineated the social responsibility of businessmen as a process regarding decision-making and action-conducting by managers based on society's expectations and value demands [30]. However, based on the theory of Neo-classical economics, Friedman opposed the viewpoint of Bowen and criticised social responsibility because he contended that opportunistic behaviours of managers to enhance their social status could potentially jeopardize shareholders' overall profits under the guise of CSR initiatives [31]. It can be concluded that the initial definitions of CSR are too narrow as they merely equated the responsibility of enterprises with the duties of businessmen. Subsequently, the essence of CSR was further expanded and polished.

Having noticed that the previous research over-emphasised enterprises' obligation and motivation while disregarding the actual performance, Carroll [32] tried to provide a more comprehensive definition of CSR. From his perspective, CSR comprises four social expectations of enterprises, which include economic, legal, ethical, and discretionary expectations. It seems that Carroll's definition provided a relatively accurate description regarding the actual condition of the contemporary business and social landscape, while one of the drawbacks of Carroll's typology of CSR is the ambiguity of the priority of those responsibilities that the enterprises should undertake. Addressing this concern, Carroll [33] introduced a pyramid structure to enhance clarity regarding the components of CSR. Notably, the discretionary responsibilities in the initial model were replaced by Philanthropic Responsibilities. In the 21st century, the concept of CSR has been extended to a more comprehensive level, reflecting an escalating social demand for enterprises. Kotler and Lee [34] described CSR as a kind of commitment that aims to improve the welfare condition of communities through the enterprises' business practices and resources. To Kramer and Porter [35], the concept of CSR is further widened and covers almost all the aspects of activities when a firm is conducting its business within a competitive environment.

## 2.2. The Relationship between CSR and CFP

The main focus of recent studies of CSR has shifted from ethical argumentations to performance studies [1,36]. In the meantime, Vogel [37] claimed that clarification of the relationship between enterprises' CSR implementation and financial performance indicates the starting of a new era of CSR study. Research on the CSR-CFP relationship was conducted as early as the 20th century when Preston and O'Bannon [38] investigated six hypotheses to identify the CSR-CFP relationship. As this research aims to probe the influence of CSR on CFP, two of the six hypotheses will be discussed as follows.

### 2.2.1. The Social Impact Hypothesis

This hypothesis is closely linked with the stakeholder theory, contending that the good practices of CSR (i.e., satisfying the reasonable demands of relevant stakeholders) will trigger a favourable CFP [38]. In their seminal research, Cornell and Shapiro [39] tried to use stakeholder theory to evaluate the enterprises' financial performance from a theoretical perspective. They found that, aside from managers and primary investors, the power of other stakeholders can impact the relationship between enterprises' strategy and their financial outcomes.

Using data from 67 U.S. companies, Preston and O'Bannon [38] unveiled a positive relationship between CSR and financial performance, and they also argued that financial performance leads to the improvement of CSR. Their findings were corroborated by Waddock and Graves [40], who used the KLD index to gauge enterprises' CSR performance and deduced that augmented financial performance results from good CSR practice. Through using a '7 + 2' criteria to evaluate the CSR performance of Chinese national state-owned enterprises (CNSOEs) under the guidance of ISO26000 [41], Zhu, Liu, and Lai [13] found that the four commendable practices (i.e., labour practices, community involvement, supply chain, and political responsibility) within the '7 + 2' criteria are positively related with the high financial performance of CNSOEs. Other studies also proved the positive and significant influence of CSR on the enterprises' financial performance based on the context of small and medium-sized enterprises (SMEs) [22], manufacturing industry [20], airlines industry [28], high-tech industry [23], etc. Based on the above discussion, the following hypothesis is established:

**Hypothesis 1a (H1a):** *The CSR initiatives of enterprises will influence their financial performance positively*.

In China, the government is one of a key participators in enterprises' CSR activities, and the imperfect market mechanisms make it even more critical for companies to establish rapport with the government to have access to some limited resources and information [42,43]. Some scholars tried to identify the function of government subsidy as a moderator in the positive CSR-CFP relationship. For instance, using Chinese Listed Companies' data from 2011 to 2017, Deng et al. [17] found that government subsidy could positively moderate the relationship between enterprises' CSR initiatives and productivity. On the contrary, Long et al. [18] reached the opposite conclusion based on similar data. However, little attention has been paid to the mediating effect of government subsidy in the CSR-CFP relationship.

Regarding the route from CSR initiatives to government subsidy, according to Shu et al. [44], firms' relatively good CSR performance helps them achieve favourable treatment and support from the government or other official institutions. Additionally, enterprises are encouraged to engage in CSR practices to increase the possibility of gaining the government subsidy, which can contribute to performance enhancement or business expansion [45,46]. On the other hand, the transmission path from government subsidy to positive firm performance also exists. In their study, Ji and Miao [47] conducted an empirical analysis of 380 Chinese manufacturing companies. They pointed out that direct and indirect government support can accelerate the transformation from CSR initiatives to enterprises' innovation performance. Through using panel data from Chinese Listed Companies, Li et al. [48]

empirically proved that the government subsidy can significantly promote the performance of circular supply chain management. Owing to the specific institutional background of China, domestic enterprises are still under the control and supervision of the Chinese government, which holds the right to dispense and collect the scarce resources of China [49,50]. If enterprises can gain access to those limited resources and get support from the Chinese government, their competitiveness within the current business environment will be enhanced tremendously [18]. Based on the above discussion, the following hypothesis is posited:

**Hypothesis 1b (H1b):** *The CSR initiatives of enterprises will influence their financial performance positively, and government subsidy will function as a mediator within that relationship.*

### 2.2.2. The Trade-Off Hypothesis

In light of the limited resources within the enterprises, they had to make various kinds of trade-offs in terms of different stakeholder groups. Contrasting the social impact hypothesis, the proponents of the trade-off hypothesis generally contend that implementing additional social responsibilities leads to extra costs for enterprises. Therefore, the surplus costs would put those socially responsible enterprises in an inferior position when compared to other relatively socially irresponsible enterprises [51–53].

Following an analysis of data from Canadian companies, Makni, Francoeur, and Bellavance [54] concluded that high CSR performance is the unidirectional 'Granger cause' of low financial performance. The findings of Barnea and Rubin [55] also proved the negative relationship between CSR and CFP because the insiders (e.g., managers and large shareholders) may try to overinvest in CSR for their interests. Using the UK-quoted companies' data, Brammer, Brooks, and Pavelin [56] embarked on an empirical investigation of CFP through stock returns. Their research assessed CSR from the points of employment, environment, and community. They concluded that the enterprises' socially responsible endeavours on the environment and community will influence their financial performance negatively. The reason why extant research shows little evidence regarding the negative relationship between CSR and CFP may be attributed to the growing awareness of social responsibilities. With the increasing attention from the public to the moral and social behaviours of enterprises, the previous factors that can undermine the CFP may be hedged. Based on the above discussion, the following hypothesis is posited:

**Hypothesis 2 (H2):** *The CSR initiatives of enterprises will influence their financial performance negatively.*

### 2.2.3. The Ambiguous Relationship

Except for the positive and negative relationship, other studies have highlighted that the linkage between CSR and CFP remains uncertain and subject to debate. As the understanding of influencing factors towards CSR and CFP become mature, McWilliams and Siegel [57] pointed out the previous econometric model used by Waddock and Graves [40] has specification error because of the omittance of Research and Development (R&D). After considering R&D factors, they empirically proved that the previously observed positive relationship between CSR-CFP became statistically insignificant. Similarly, Surroca, Tribó, and Waddock [58] focused on the role of intangible resources (e.g., human resources, innovative capability, etc.) in terms of the CSR-CFP relationship, and they claimed that there is no direct relationship between CSR and CFP because of the full mediation effect of the intangible resources. McWilliams and Siegel [59] used the supply and demand side of CSR attributes as a relatively unique perspective to explore the impact of CSR on CFP. Despite a lack of empirical data, they argued that the relationship between CSR and CFP is neutral. Additionally, Mahoney and Roberts [60] tried to use both comprehensive and separate measures of CSR and argued that the good practice of CSR can not result in high financial performance.

In a comprehensive literature review encompassing 53 relevant papers, Coelho, Jayantilal, and Ferreira [3] identified that the relationship between enterprises' CSR implementation and their CFP remains elusive. There are multiple types regarding the link between CSR and CFP, including positive, negative, neutral, direct and positive, indirect and positive, etc. In general, a larger proportion of the literature still upholds that CFP is positively related to CSR, and that kind of relationship would become tighter accompanying the improvement of an enterprise's Environmental, Social, and Governance (ESG) scores. Additionally, through a comprehensive analysis of the articles that probe the relationship between CSR and CFP, Griffin and Mahon [61] found that 33 of them verified the positive relationship between CSR and CFP; 19 of them concluded that the implementation of CSR would undermine the financial performance of enterprises; and 9 of them failed to establish a clear connection. Combining the findings of prior research and considering the finite resources of enterprises, we hypothesise that a firm's consistent adherence to CSR practices may not always yield a straightforward positive or negative impact on its financial performance. Instead, it may demonstrate a non-linear relationship. At the initial stage, the adoption of CSR measures may lead to a favourable impact on the financial performance of the enterprise, attributed to the establishment of a reputation as 'good citizens' and the ensuing support from the public. However, the continuous implementation of CSR initiatives may have a 'crowding out effect' on business resources that could otherwise be allocated elsewhere and therefore may weaken the enterprise's financial performance to some extent. Based on the above discussion, the following hypothesis is established:

**Hypothesis 3 (H3):** *There is a non-linear relationship between CSR and CFP.*

## 3. Research Methodology

### 3.1. Measures

#### 3.1.1. Corporate Financial Performance

Recent research used two types of measurement as the proxy for CFP: market-based and accounting-based. The market-based measures are closely related to an enterprise's stock performance, and the main indicators commonly used in research are price per share [29,62], price-to-book value [56], share price appreciation [29], etc. In a similar vein, the accounting-based measures are represented by multiple kinds of financial indicators, such as return on assets (ROA), return on equity (ROE), return on sales (ROS), profit margin, etc. [9,40,52–54]. Davidson and Worrell [63] favoured market-based measures because they claimed that those measures are closely related to the direct stakeholders (i.e., shareholders). However, some scholars held the opposite view [53,55,64] as they believed that accounting-based measures can better capture the unique characteristics of enterprises than market-based measures. Considering the availability and advantage of data, this study will opt for accounting-based measures, ROA, as the proxy for CFP.

#### 3.1.2. Corporate Social Responsibility

Based on the stakeholder theory, enterprises' CSR can be categorised into responsibility to shareholders, creditors, customers, suppliers, employees, government, and communities [65]. As the owners of enterprises, shareholders will grow with them, and the financial performance is of critical importance to those shareholders. Creditors are concerned with a firm's financial performance because it is directly related to the level of debt it has to incur. Customers are the receivers of enterprises' products or services, and the financial performance of the enterprises affects their survival, which in turn influences whether loyal customers can buy the goods or services from their favourite enterprises. Considering suppliers are situated in the upper stream of enterprises' value chain, the enterprises' financial performance will affect the relationship's stability with their suppliers [66]. By implementing enterprises' CSR, employees' confidence towards the enterprises can help them better achieve their financial goals [17]. As the regulator of the enterprises, the responsibility to the government is related to the normal operation of enterprises and, therefore,

affects their financial performance. Enterprises' active fulfilment of social responsibility to the communities can help them to yield positive reputational effects, which indirectly affects their financial performance [67]. In short, it can be seen that the enterprises' CSR to different stakeholders and the corresponding financial performance are closely related.

According to what has been discussed above and the Stakeholder Theory, this study will choose multiple indicators from the perspective of different stakeholders as the proxy of enterprises' CSR performance [68–70].

(1)    Responsibility to shareholders (*SHR*)

This study chooses ***Earnings per share growth rate*** to represent the enterprises' responsibility to their shareholders. According to Zhao et al. [67], the main responsibility of enterprises to their shareholders is to guarantee continuous profit to them. Considering that this indicator is composed of the earnings for shareholders, it serves as a suitable metric to gauge the extent of an enterprise's CSR towards its shareholders. The higher this rate is, the higher the economic responsibility undertaken by the enterprises to their shareholders [69].

(2)    Responsibility to creditors (CRR)

***Current Ratio*** will become the proxy for the responsibility to creditors of enterprises. The main responsibility of enterprises to their creditors is reflected in the ability to repay loans on time [66]. Since this indicator is composed of assets and liabilities, a higher ***Current Ratio*** means that the enterprises have more funds to repay their loans to creditors.

(3)    Responsibility to customers (CUR)

A low cost of business means that the quality of enterprises' products or services will be affected to some extent, and therefore the customers' rights will be jeopardised [67]. Thus, ***Cost of main business ratio*** will represent the enterprises' responsibility to their customers. This indicator is measured by business costs and revenues. The high level of this ratio indicates that the costs of enterprises will exceed their revenues. Thus, the customers' rights and interests are more likely to be safeguarded.

(4)    Responsibility to suppliers (SUR)

The social responsibility of enterprises towards their suppliers should be the timely payment for goods and services rendered [67]. ***Accounts Payable Turnover Ratio*** will be the proxy for enterprises' responsibility to their suppliers. A higher value of this ratio indicates that enterprises are making payments to suppliers in a shorter time frame, ensuring that suppliers' funds are not unduly tied up, which reflects a proactive responsibility fulfilling to their suppliers.

(5)    Responsibility to employees (EMR)

Wage and welfare are the primary concerns of employees in an enterprise, and it is also a key CSR issue [67,71]. Therefore, the enterprises' responsibility to their employees can be reflected in and statistically measured by the wages and welfare of the employees. The ***Employee Profitability Level*** is used to quantify the welfare level of employees, and a higher level of this indicator signifies that employees can gain more benefits from the companies. Accordingly, a high level of this indicator also means that a company is fulfilling more economic and legal responsibilities towards its employees.

(6)    Responsibility to government (GOR)

According to Xin [72], paying taxes and complying with laws and regulations are two of the social responsibilities that enterprises need to assume to the government. To quantify that kind of responsibility, this study chooses ***Asset Tax Rate*** as the proxy for enterprises' responsibility towards the government. A high level of this rate indicates that enterprises will be active in paying taxes to the government; therefore, a high level of responsibility to the government can be reflected.

(7)    Responsibility to communities (CMR)

As indicated by Cramer [73], enterprises' social responsibility to the local communities can be partly reflected in the financial assistance and donations to those communities. ***Social donation expenditure rate*** is used to demonstrate the costs of enterprises to the maintenance and development of society and communities. The higher this rate is, the higher the enterprises will pay from their total revenue. Therefore, enterprises with a relatively high social donation expenditure rate can be regarded as pro-social enterprises.

### 3.1.3. Mediating Variable and Overall CSR Gauge

To confirm whether the government subsidy could function as a potential mediator within the CSR-CFP relationship, this study will use the real amount of government subsidy (GS) from the CSMAR database [18,47]. Additionally, to facilitate the testing of the mediation effect and the subsequent threshold effect, the author will construct an indicator (CSRCI) to gauge the overall CSR performance of the enterprises. In his seminal study, Carroll [33] set the weight of economic, legal, ethical, and philanthropic responsibility as 4, 3, 2 and 1, respectively. Following the framework of Carroll, Zhang et al. [69] argued that the expenditures on different stakeholders should be allocated separately by a weight of 0.6 and 0.4 between internal and external stakeholders. In the meantime, all aspects of responsibility should be distributed among internal and external stakeholders equally (i.e., an allocation weight of 0.5). Considering the nature of legal responsibility, it will be equally distributed among all the stakeholders. Therefore, the final weight for shareholders, creditors, customers, suppliers, employees, government, and communities will be 28.29%, 12.29%, 18.29%, 8.29%, 22.29%, 4.29%, 6.26%, respectively (see Figure 1).

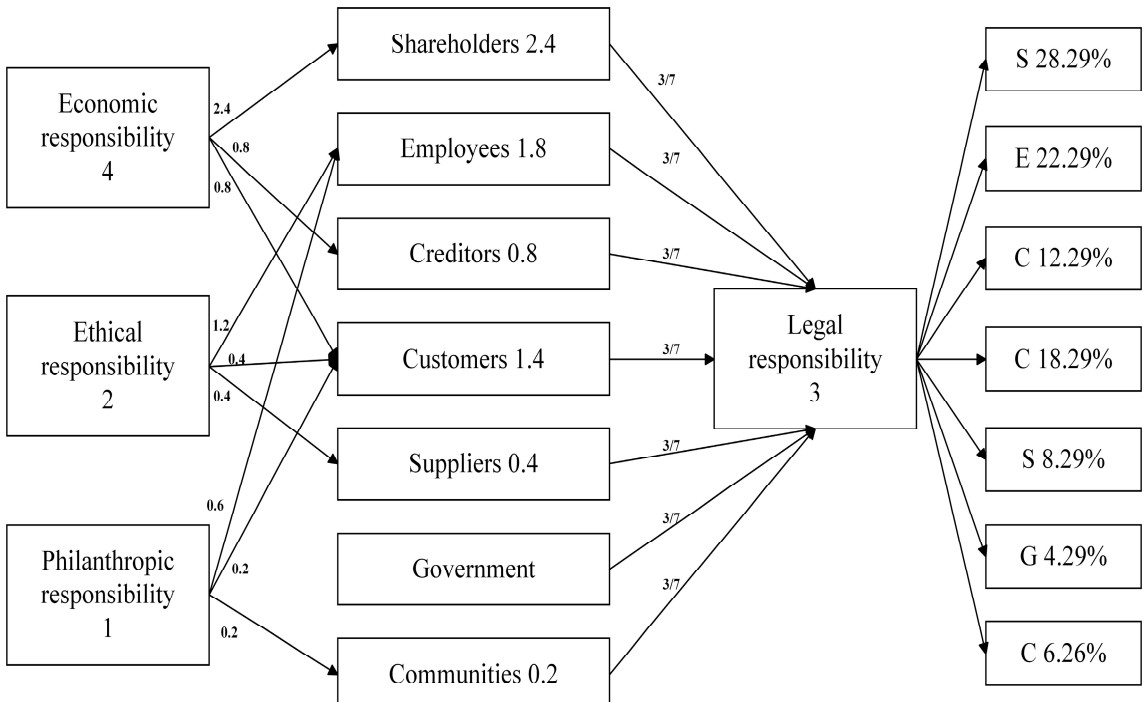

**Figure 1.** Weight distribution of different stakeholders.

### 3.1.4. Control Variables

Following the previous literature [21,58], a set of control variables are also included in the regression. *Firm size* (SIZE) is the natural logarithm of total assets. Additionally, the intensity of R&D may also influence both the CSR implementation and enterprises' financial performance [57]. Generally, higher R&D expenditures tend to be associated with better financial performance. Therefore, this study will also include the *intensity of R&D* (RDI) as a control variable.

The summary and calculation method of the variables mentioned in this research can be found in Table 1.

**Table 1.** Summary of the definition of relevant variables.

| Type | Code | Name | Indicator | Calculation Method |
|---|---|---|---|---|
| Dependent Variable | CFP | Corporate financial performance | Return on assets | Net income/Total assets |
| Independent Variables | SHR | Responsibility to shareholders | Earnings per share growth rate | (Earnings per share for current period-Earnings per share for last period)/Earnings per share for last period |
| | CRR | Responsibility to creditors | Current ratio | Current assets/Current liabilities |
| | CUR | Responsibility to customers | Cost of main business ratio | Main business cost/Main business income |
| | SUR | Responsibility to suppliers | Accounts payable turnover ratio | Main business cost/Accounts payable |
| | EMR | Responsibility to employees | Employee profitability level | Employees' salary/Net revenue |
| | GOR | Responsibility to government | Asset tax rate | Total taxes/Total assets |
| | CMR | Responsibility to communities | Social donation expenditure rate | Social donation expenditure/Main business income |
| Mediating Variable | GS | Government subsidy | / | / |
| Threshold Variable | CSRCI | CSR comprehensive index | The weighted score of different stakeholders | / |
| Control Variables | SIZE | Firm size | / | Natural logarithm of total assets |
| | RDI | Intensity of research and development | / | Expenditure on R&D/Main business income |

### 3.2. Data Collection

The sample of this research includes all the A-shares companies listed in the Shanghai and Shenzhen stock exchanges from 2017 to 2022. The data are collected from the China Stock Market and Accounting Research (CSMAR) database, which covers almost all the potential objective indicators of Chinese Listed Companies, and this database has been proven to be efficient by many studies [17,18,74]. Considering the particularity of the capital structure and accounting measurement, the financial companies and insurance companies are excluded from the whole sample. Moreover, the observations labelled with ST and ST* are omitted from the sample as they are in unusual conditions. To mitigate the influence of outliers, all the continuous variables are winsorized at the top and bottom of 1%. Finally, when all the incomplete observations are dropped, a final sample including 4958 listed enterprises composed of 27,684 firm–year observations is confirmed.

The Hausman test was generally adopted to confirm the choice of random-effects or fixed-effects mode. If the *p*-value of the Hausman test is no more than 0.05, the estimates of the fixed-effects model will be consistent and more appropriate [75,76]. Based on the result of the Hausman test, this study will adopt the panel data regression model using fixed effects as the baseline model. Following the previous studies [25,77], the firm fixed effect are included in this study to control the unobservable and time-invariant heterogeneity. The author also takes the time effect into the model specification, which aims to form a two-way fixed effects model and control other timely trends.

According to the given hypothesis and theoretical assumption, the panel data model of this study is as follows:

$$CFP_{i,t} = \alpha_0 + \alpha_1 SHR_{i,t} + \alpha_2 CRR_{i,t} + \alpha_3 CUR_{i,t} + \alpha_4 SUR_{i,t} + \alpha_5 EMR_{i,t} + \alpha_6 GOR_{i,t} + \\ \alpha_7 CMR_{i,t} + \sum_{j=1}^{J} \alpha_i Controls_{i,j} + \varepsilon_{i,t} \qquad (1)$$

where $\alpha_0$ is the intercept, $SHR_{i,t}$ represents enterprise $i$'s responsibility to its shareholders at time $t$, $Controls_{i,j}$ is the $j$th control variable for firm $i$, $j = 1, 2, \ldots, J$, and $\varepsilon_{i,t}$ is the random error term.

To test the existence of a non-linear relationship within the CSR-CFP relationship, this study adopts the econometric model issued by Hansen [78] to find the threshold effect:

$$CFP_{i,t} = \alpha_0 + \alpha_1 CSRCI_{i,t} \cdot I(CSR_{i,t} \le \mu) + \alpha_2 CSRCI_{i,t} \cdot I(CSR_{i,t} > \mu) + \sum_{j=1}^{J} \alpha_i Controls_{i,j} + \varepsilon_{i,t} \qquad (2)$$

where $\alpha_0$ is the intercept, $CSR_{i,t}$ is the threshold variable, $I(\bullet)$ is the indicator function, $Controls_{i,j}$ is the $j$th control variable for firm $i$, $j = 1, 2, \ldots, J$, and $\varepsilon_{i,t}$ is the random error term.

## 4. Results

### 4.1. Descriptive Statistics

The summary statistics of relevant variables is shown in Table 2. As indicated by the table, the mean value financial performance (ROA) of the whole sample is 4.52%, which also shows the average asset utilisation efficiency of Chinese A-share Listed Companies. Regarding the status of CSR implementation, the table shows that the average value of (CSRCI) is 0.923, ranging from −11.192 to 9.715. The broad spectrum of distribution provides clues into the capacity and willingness of Chinese companies regarding CSR implementation. Some firms still neglect the significance of CSR, which can be especially reflected in the responsibility to shareholders (SHR) (the minimum value is −40 with a mean value of −0.53).

**Table 2.** Descriptive statistics of relevant variables.

|        | N       | SD        | Mean       | Min      | Max        | Median   |
|--------|---------|-----------|------------|----------|------------|----------|
| ROA    | 27,684  | 0.089     | 0.0452     | −0.3961  | 0.2717     | 0.0463   |
| SHR    | 27,683  | 5.3057    | −0.5303    | −40      | 14.129     | 0        |
| CRR    | 27,685  | 2.4839    | 2.6013     | 0.3325   | 15.6897    | 1.7874   |
| CUR    | 27,684  | 0.1872    | 0.6836     | 0.1169   | 1.0139     | 0.7164   |
| SUR    | 27,664  | 10.8186   | 7.3111     | 0.4405   | 79.6351    | 4.2417   |
| EMR    | 23,208  | 0.0146    | 0.0062     | −0.0282  | 0.0925     | 0.0023   |
| GOR    | 23,217  | 0.0053    | 0.0050     | 0.0003   | 0.0386     | 0.0037   |
| CMR    | 15,334  | 0.001     | 0.0005     | 0        | 0.0068     | 0.0001   |
| CSRCI  | 15,331  | 2.0825    | 0.9231     | −11.1922 | 9.7152     | 0.7888   |
| GS     | 23,045  | 5027.3255 | 1731.1160  | 0        | 37,836.891 | 202.71   |
| SIZE   | 23,217  | 1.31      | 22.2550    | 19.8569  | 26.3729    | 22.0545  |
| RDI    | 21,085  | 0.0564    | 0.0547     | 0.0002   | 0.3292     | 0.04     |

The correlation coefficients among the major variables are shown in Table 3. The linear dependency between two variables can be measured through Pearson's correlation coefficient (*r*). Whether from a general (CSRCI) or a separate (SHR, CRR, CUR, SUR, EMR, GOR, CMR) perspective, the relevant measurements of enterprises' CSR implementation significantly correlate with their financial performance. Most of the correlation coefficients are positive, which can provide preliminary evidence that *H1a* is reliable. Additionally, the significance of correlation coefficients between control variables and the majority of other variables can demonstrate the choice of control variables is reasonable. According to Myers [79] and Hasan et al. [77], once the Variance Inflation Factor (VIF) is lower than 10, the problem of multicollinearity is not severe. After running the OLS regression, an average VIF of 1.22 shows that multicollinearity is not a major concern in this research.

**Table 3.** Correlation coefficient matrix.

| | ROA | SHR | CRR | CUR | SUR | EMR | GOR | CMR | CSRCI | GS1 | SIZE | RDI |
|---|---|---|---|---|---|---|---|---|---|---|---|---|
| ROA | 1 | 0.18 *** | 0.42 *** | −0.45 *** | 0.17 *** | 0.15 *** | 0.23 *** | 0.11 *** | 0.33 *** | 0.00 | −0.12 *** | 0.07 *** |
| SHR | 0.35 *** | 1 | 0.08 *** | −0.05 *** | −0.09 *** | 0.05 *** | −0.00 | −0.02 ** | 0.56 *** | 0.02 ** | −0.04 *** | 0.06 *** |
| CRR | 0.23 *** | 0.05 *** | 1 | −0.43 *** | 0.11 *** | 0.24 *** | 0.01 | 0.12 *** | 0.41 *** | −0.10 *** | −0.48 *** | 0.34 *** |
| CUR | −0.34 *** | −0.06 *** | −0.39 *** | 1 | 0.23 *** | −0.15 *** | −0.12 *** | −0.27 *** | 0.01 | 0.02 ** | 0.24 *** | −0.40 *** |
| SUR | 0.04 *** | −0.00 | 0.09 *** | 0.21 *** | 1 | 0.03 *** | 0.21 *** | −0.04 *** | 0.55 *** | −0.04 *** | −0.03 *** | −0.29 *** |
| EMR | 0.03 *** | 0.10 *** | 0.07 *** | −0.02 ** | −0.01 | 1 | −0.01 | 0.06 *** | 0.15 *** | −0.09 *** | −0.41 *** | 0.18 *** |
| GOR | 0.13 *** | 0.03 *** | −0.07 *** | −0.15 *** | 0.03 *** | −0.05 *** | 1 | 0.04 *** | 0.08 *** | −0.02 ** | −0.04 *** | −0.16 *** |
| CMR | 0.02 *** | −0.01 | 0.12 *** | −0.27 *** | −0.04 *** | 0.01 | 0.03 *** | 1 | −0.03 *** | 0.02 ** | −0.06 *** | 0.08 *** |
| CSRCI | 0.32 *** | 0.85 *** | 0.23 *** | −0.00 | 0.46 *** | 0.09 *** | 0.02 *** | −0.01 | 1 | −0.05 *** | −0.17 *** | −0.06 *** |
| GS | 0.03 *** | 0.02 *** | −0.11 *** | 0.06 *** | 0.00 | −0.09 *** | 0.07 *** | −0.02 ** | 0.00 | 1 | 0.28 *** | 0.04 *** |
| SIZE | −0.01 | 0.01 | −0.36 *** | 0.22 *** | 0.00 | −0.27 *** | 0.12 *** | −0.06 *** | −0.04 *** | 0.42 *** | 1 | −0.34 *** |
| RDI | −0.07 *** | −0.00 | 0.28 *** | −0.44 *** | −0.15 *** | 0.07 *** | −0.17 *** | 0.12 *** | −0.02 *** | 0.00 | −0.25 *** | 1 |

Note: lower-triangular cells report Pearson's correlation coefficients, and upper-triangular cells are Spearman's rank correlation. ** $p < 0.05$, *** $p < 0.01$. SHR = responsibility to shareholders; CRR = responsibility to creditors; CUR = responsibility to customers; SUR = responsibility to suppliers; EMR = responsibility to employees; GOR = responsibility to government; CMR = responsibility to communities; CSRCI = CSR comprehensive index; GS = government subsidy; SIZE = firm size; RDI = intensity of R&D.

### 4.2. Testing of Hypotheses

The results of fixed-effects regression are shown in Table 4, where column 1 shows the regression results without control variables, columns 2–3 demonstrate the gradual incorporation of control variables into the equation, and column 4 aims to illustrate the overall influence of different perspectives of stakeholders on enterprises' financial performance.

**Table 4.** Fixed effects regression analysis.

| | (1) | (2) | (3) | (4) |
|---|---|---|---|---|
| | ROA | ROA | ROA | ROA |
| SHR | 0.003 *** | 0.003 *** | 0.003 *** | |
| | (36.063) | (36.256) | (35.691) | |
| CRR | 0.002 *** | 0.003 *** | 0.002 *** | |
| | (3.694) | (5.302) | (2.956) | |
| CUR | −0.399 *** | −0.384 *** | −0.428 *** | |
| | (−46.030) | (−44.369) | (−45.670) | |
| SUR | 0.001 *** | 0.001 *** | 0.001 *** | |
| | (12.634) | (12.568) | (9.704) | |
| EMR | 0.326 *** | 0.343 *** | 0.316 *** | |
| | (7.520) | (7.998) | (7.286) | |
| GOR | 0.639 *** | 1.332 *** | 1.346 *** | |
| | (2.654) | (5.471) | (4.649) | |
| CMR | −3.995 *** | −3.797 *** | −3.423 *** | |
| | (−5.719) | (−5.485) | (−4.843) | |
| CSRCI | | | | 0.011 *** |
| | | | | (34.486) |
| SIZE | | 0.030 *** | 0.023 *** | 0.028 *** |
| | | (14.406) | (10.876) | (12.362) |
| RDI | | | −0.628 *** | −0.724 *** |
| | | | (−22.139) | (−23.033) |
| _cons | 0.305 *** | −0.368 *** | −0.155 *** | −0.546 *** |
| | (46.422) | (−7.806) | (−3.200) | (−10.789) |
| Year fixed effects | Yes | Yes | Yes | Yes |
| Firm fixed effects | Yes | Yes | Yes | Yes |
| N | 15,331 | 15,331 | 14,057 | 14,057 |
| r2 | 0.282 | 0.295 | 0.349 | 0.186 |
| *p* | 0.000 | 0.000 | 0.000 | 0.000 |
| F | 367.680 | 361.593 | 390.522 | 292.175 |

Note: t-value in parentheses; *** $p < 0.01$.

According to column 3, all the stakeholders' perspectives have a significant impact ($p < 0.01$) towards enterprises' ROA. Except for the factors of customers (CUR) ($\beta = -0.428$)

and communities (CMR) ($\beta = -3.423$), other aspects of CSR implementation can exert a positive and significant influence on enterprises' financial performance, which can partly support the rationale of *H1a*. In column 4, ROA is related to a comprehensive index of CSR implementation (CSRCI) with a set of control variables under the control of the firm and time fixed effects. The result reveals that CSR implementation is positively and statistically significant to financial performance at the significance level of 1%. Therefore, we can safely conclude that *H1a* is accepted while *H2* is rejected by this research. This finding corroborates the research of Zhu, Liu, and Lai [13] while contradicting that of Brammer, Brooks, and Pavelin [56], providing further evidence that good CSR initiatives have the great potential to serve as the 'activation trigger' of firms' enhanced financial performance.

*4.3. Testing of Mediation Effect*

To determine whether the government subsidy can function as a conduction mechanism from CSR implementation to the enhanced enterprises' financial performance, the author conducts a three-step hierarchical regression. The results are reported in columns 1–3 of Table 5. According to the extant research [77,80], the existence of the mediating role of government subsidy (GS) within the CSR-CFP relationship must satisfy the following conditions: the coefficients of the effect of CSRCI on the dependent variable (i.e., ROA) and the mediator (i.e., GS) should be significant. Additionally, after controlling CSRCI, GS needs to impact ROA significantly, and the main effect of CSRCI should be undermined by the mediation effect. In column 1, CSRCI has a positive and significant influence on ROA, which is consistent with the prior test. However, the results from columns 2 and 3 reveal that both the routes of CSRCI-GS and GS-ROA are not significant. Therefore, this study can not confirm the mediating role of government subsidy within the CSR-CFP relationship, and *H1b* is rejected.

**Table 5.** CSR and CFP: the mediating role of GS.

|  | (1) | (2) | (3) |
|---|---|---|---|
|  | **ROA** | **GS** | **ROA** |
| CSRCI | 0.011 *** | 7.363 | 0.011 *** |
|  | (34.486) | (0.359) | (34.401) |
| SIZE | 0.028 *** | 1316.413 *** | 0.028 *** |
|  | (12.362) | (8.952) | (12.218) |
| RDI | −0.724 *** | 3272.657 | −0.725 *** |
|  | (−23.033) | (1.620) | (−23.006) |
| GS |  |  | 0.000 |
|  |  |  | (0.837) |
| _cons | −0.546 *** | $-2.81 \times 10^4$ *** | −0.543 *** |
|  | (−10.789) | (−8.619) | (−10.656) |
| Year fixed effects | Yes | Yes | Yes |
| Firm fixed effects | Yes | Yes | Yes |
| N | 14,057 | 14,006 | 14,006 |
| r2 | 0.186 | 0.052 | 0.187 |
| *p* | 0.000 | 0.000 | 0.000 |
| F | 292.175 | 69.812 | 259.147 |

Note: t-value in parentheses; *** $p < 0.01$.

*4.4. Robustness Check*

In the previous parts, ROA is used as an indicator of enterprises' financial performance. At the same time, ROE is another commonly used variable that can measure financial performance in the literature and has been proven to be valid by many scholars [20,21,60,81]. In order to increase the robustness of the outcome, we use ROE as a substitutive proxy of enterprises' financial performance, and the results are shown in Table 6.

**Table 6.** Dependent variable substitution test.

| | (1) | (2) | (3) |
|---|---|---|---|
| | ROE | ROE | ROA-Base |
| SHR | 0.007 *** | | |
| | (28.927) | | |
| CRR | −0.001 | | |
| | (−0.610) | | |
| CUR | −0.911 *** | | |
| | (−38.645) | | |
| SUR | 0.002 *** | | |
| | (6.638) | | |
| EMR | 0.934 *** | | |
| | (8.564) | | |
| GOR | 2.100 *** | | |
| | (2.884) | | |
| CMR | −9.229 *** | | |
| | (−5.192) | | |
| CSRCI | | 0.022 *** | 0.011 *** |
| | | (28.846) | (34.486) |
| SIZE | 0.036 *** | 0.050 *** | 0.028 *** |
| | (6.853) | (8.919) | (12.362) |
| RDI | −1.268 *** | −1.460 *** | −0.724 *** |
| | (−17.771) | (−19.017) | (−23.033) |
| _cons | −0.071 | −0.968 *** | −0.546 *** |
| | (−0.584) | (−7.830) | (−10.789) |
| Year fixed effects | Yes | Yes | Yes |
| Firm fixed effects | Yes | Yes | Yes |
| N | 14,057 | 14,057 | 14,057 |
| r2 | 0.264 | 0.133 | 0.186 |
| p | 0.000 | 0.000 | 0.000 |
| F | 261.506 | 195.118 | 292.175 |

Note: t-value in parentheses; *** $p < 0.01$.

Compared with the baseline regression (see Table 4), neither the significance of coefficients nor the direction of effects change significantly. The results from columns 2 and 3 indicate that the overall CSR initiatives of Chinese A-share Listed Companies can positively and significantly influence their financial performance from the perspective of ROE ($\beta =$ 0.022, $p < 0.01$). Again, **H1a** can be confirmed by this study.

*4.5. Testing of Threshold Effect*

In the initial stage, the financial performance of enterprises may increase as they gain public support through conducting CSR initiatives. However, the continuous implementation of CSR will largely consume the firms' resources, which may trigger an uncertain route for their future development [82]. To investigate whether there is not just a simple positive or negative relationship between CSR and CFP but a non-linear one, this study will adopt the threshold regression method [78] to test **H3**.

The results of the threshold effects test as shown in Table 7 demonstrate that both the single threshold and double threshold are significant at the significance level of 1%. In contrast, the triple threshold can not pass the significance test ($p > 0.5$). Thus, we can safely conclude that there is a double threshold effect within the CSR-CFP relationship. Next, the original single threshold model should be modified as a double threshold model as follows:

$$CFP_{i,t} = \alpha_0 + \alpha_1 CSR_{i,t} \cdot I(CSR_{i,t} \leq \mu_1) + \alpha_2 CSR_{i,t} \cdot I(\mu_1 < CSR_{i,t} \leq \mu_2) + \alpha_3 CSR_{i,t} \cdot I(CSR_{i,t} > \mu_2) + \sum_{j=1}^{J} \alpha_i Controls_{i,j} + \varepsilon_{i,t}$$

(3)

**Table 7.** Threshold effect test.

| Type of Threshold | Bootstrap Times | F-Value | p-Value | Critical Value | | |
|---|---|---|---|---|---|---|
| | | | | 10% | 5% | 1% |
| Single | 300 | 158.23 | 0.000 | 7.416 | 9.624 | 13.377 |
| Double | 300 | 44.850 | 0.000 | 9.505 | 10.825 | 17.680 |
| Triple | 300 | 16.960 | 0.410 | 29.784 | 34.666 | 40.484 |

After running the threshold value estimation process (see Table 8), the two threshold values are −8.208 and 2.687, and the corresponding 95% confidence intervals are [−11.072, −7.210] and [2.637, 2.742]. To further substantiate the veracity of the threshold estimates, a likelihood ratio (LR) function graph is plotted (see Figure 2). The LR statistics for both two threshold values lie below the LR benchmark value of 7.35 at the significance level of 5%, which reveals that the estimated threshold values are equal with the authentic threshold values [83]. The null hypothesis of the existence of a double threshold is therefore accepted and *H3* is also supported by this research.

**Table 8.** Estimation of threshold value and confidence interval.

| Type of Threshold | Threshold Value | 95% Confidence Interval |
|---|---|---|
| First threshold | −8.208 | [−11.072, −7.210] |
| Second threshold | 2.687 | [2.637, 2.742] |

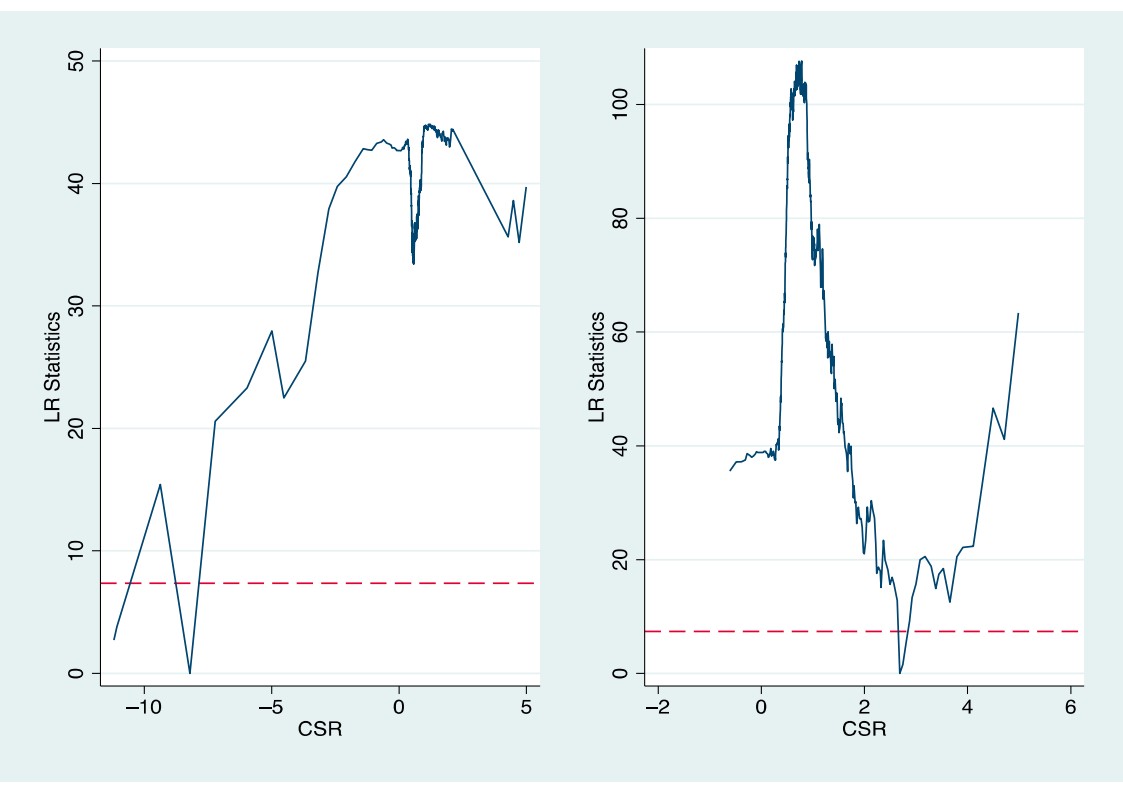

**Figure 2.** Threshold value likelihood ratio function graph.

Based on the two estimated threshold values, a threshold regression analysis (see Table 9) is used to delineate the non-linear relationship between CSR and CFP in detail. Generally, regardless of the intervals into which overall CSR implementations fall, the general relationship between CSR and CFP is still positive at the significance level of 1%, which further verifies the reliability of *H1*. Specifically, before the CSR implementation

level reaches the first threshold, the coefficient is 0.012 ($p < 0.01$), which means holding other variables constant, the change of a single unit in CSR implementation will trigger an increase of 0.012 units in corresponding financial performance. When the overall CSR implementation is located between the first and the second threshold, for every unit increase in CSR implementation, the financial performance will increase by 0.018 units when holding other factors constant ($p < 0.01$). By the same token, after the CSR fulfilment level jumps over the second threshold, controlling for the constancy of other variables, an increase of one unit in CSR implementation will cause 0.006 units of increase in financial performance.

**Table 9.** Threshold regression analysis.

|  | (1) |
| --- | --- |
|  | **ROA** |
| SIZE | 0.015 *** |
|  | (6.422) |
| RDI | −0.834 *** |
|  | (−21.905) |
| CSRCI (CSRCI ≤ −8.208) | 0.012 *** |
|  | (24.753) |
| CSRCI (−8.208 < CSRCI ≤ 2.687) | 0.018 *** |
|  | (22.697) |
| CSRCI (CSRCI > 2.687) | 0.006 *** |
|  | (8.716) |
| _cons | −0.263 *** |
|  | (−5.141) |
| r2 | 0.200 |
| F | 374.200 |

Note: t-value in parentheses; *** $p < 0.01$.

In a nutshell, although the overall influence of CSR implementation on enterprises' financial performance is positive, the impact level fluctuates with the level of CSR implementation, showing a flat-steep-flat trend. Thus, the positive relationship between CSR and CFP is not purely linear, and *H3* is proven again.

## 5. Discussion

### 5.1. Theoretical Implications

The findings of this research are expected to extend the extant literature regarding the enterprises' CSR implementation and government subsidy. First, as a comparison with many prior studies which merely use the single indicator as the proxy for CSR performance of enterprises [55,57,60,84], this research tries to measure CSR from different stakeholders' perspective. By decomposing the single total effect into sub-effects aligned with seven stakeholders dimensions, this research robustly demonstrates the function of enterprises' CSR implementation to their financial performance. While a plethora of prior studies have suggested that the relationship between CSR and CFP is negative or even irrelevant [40,56,58], this study effectively illustrates the positive CSR-CFP relationship by focusing on China as an emerging economy and utilising panel data and various econometric methods. In addition, by incorporating objective financial data, this study overcomes the inconsistency arising from the use of questionnaires or subjective evaluations in previous research [22], and further sheds light on stakeholder theory and the social impact hypothesis in guiding the management and operation of modern enterprises.

Second, despite some prior studies having shown indications that government intervention can function as a moderator within the CSR-CFP relationship [17,18], this study aims to empirically probe the mediating role of government intervention in the CSR-CFP relationship by using the indicator of government subsidy. However, the results from the three-step regression analysis fail to support the existence of a mediation effect, which contradicts the notion that government intervention can always trigger a positive impact

regarding the long-term financial performance of enterprises [85,86]. As for the reason why the mediation effect can not be established, two possibilities arise. On the one hand, due to the intricacies of China's political system, the government and other relevant authorities might encounter challenges in assessing the enterprises' CSR performance. Moreover, the hierarchical management system could impede the conduction route of CSR initiatives to government subsidy. In addition, the current subsidy policy may not encompass all CSR dimensions, thus leaving certain CSR initiatives undertaken by companies outside the policy framework ineligible for corresponding government subsidy. On the other hand, Chinese listed companies have a relatively large system and complex businesses, and government subsidy may not wield sufficient influence to significantly and positively impact their financial performance. Moreover, considering the existence of managerial opportunism [3,87], it is still questionable whether the government subsidy is appropriately utilised as a resource for corporate development. Nonetheless, considering that the government is a critical intermediary [17] between businesses and the public, its influence on firms warrants further study.

Third, the non-linear relationship within CSR-CFP indicates that the traditional viewpoint that simply treats the relationship between the two of them as linear may have some flaws. Factors such as enterprises' reputation [62], productivity [77], word-of-mouth from customers [88] as noted in prior studies, along with the mediating role of government subsidy that this study endeavours to examine, may all have the potential to complicate the relationship between CSR and CFP further. Nevertheless, few scholars have ventured to test the possibility of a non-linear relationship between CSR and CFP, as well as the underlying mechanisms that give rise to such a phenomenon. While the six hypotheses introduced by Preston and O'Bannon [38], which underscore the reciprocal interaction between CSR and CFP, retain their logical foundation and validity, they are predicted on the assumption of a linear relationship. Following the results of this empirical study, the contribution of CSR to enterprises' financial performance will fluctuate with the level of CSR implementation. However, the existing theories regarding the relationship between CSR and CFP do not explain the potential non-linear relationship between the two. It can be seen that contemporary business practices urgently require more innovative and rational theoretical foundations.

### 5.2. Managerial Implications

First, based on the threshold regression method, this study finds that CSR can not optimally contribute to CFP when the CSR implementation level is insufficient or excessive. An excess of CSR endeavours may cause a 'Crowding Out Effect' on enterprise resources. On the contrary, if firms do not actively fulfil their CSR, employees will harbour scepticism and exhibit low labour productivity, which is not conducive to enterprises' sustainable development [17]. Many previous studies have advocated that enterprises should make every effort to engage in CSR activities to bolster their legitimacy and retain their employees [89,90], but they neglected that profit maximisation is the fundamental pursuit of enterprises [4]. When enterprises strike a balance in the level of CSR implementation, positioned within the intermediate range, the momentum for enhancing financial performance attains its zenith. As a result, firms are more motivated to fulfil their CSR obligations. Although enterprises are expected to be 'good citizens' and maximise their CSR performance, moderate CSR performance may be a win-win choice for the long-term development of both society and the enterprises themselves.

Second, several previous studies have underscored the pivotal role of government in enhancing the financial performance of enterprises [91,92]. Nevertheless, in their seminal study, Allen, Qian, and Qian [93] posited that enterprises that obtain a government subsidy will perform badly, as the main stakeholders might perceive that the purpose of enterprises' CSR initiatives is designed to meet political requirements rather than authentic social care. This study, however, does not yield conclusive evidence on the potential of government subsidy to bridge the gap between CSR and CFP. However, that does not negate the gov-

ernment's potential to coordinate between society and enterprises. In order to better fulfil their responsibilities and promote the healthy development of enterprises, the government and relevant authorities can contemplate expanding communication channels between enterprises and higher management organizations. Concurrently, the relevant rules and regulations should be further improved to ensure the support and subsidy received by enterprises can be genuinely utilised for the development of enterprises.

*5.3. Limitations and Future Research*

Through using the panel data of Chinese Listed Companies, this study empirically probes the potential relationship between enterprises' CSR implementation and their corresponding financial performance. It is undeniable that this study has some limitations and deserves further improvement in the future. First, the research perspective of this study is placed on the listed companies in China. However, the role of SMEs in the business environment can not be neglected. Thus, those SMEs can also be included in future research regarding CSR and CFP. In addition, this study merely probes the relationship between CSR and enterprises' financial performance, which is not the sole indicator of enterprises' performance. In the future, other perspectives, for example, the enterprises' social performance, can also be correlated with enterprises' social responsibility. Third, it is noteworthy that this study adopts a set of relatively new variables to explore the relationship between CSR and CFP. Although the positive relationship has been supported by the empirical findings using numerical data from Chinese Listed Companies, the selection of new variables may raise issues of comparability and necessitates further empirical and practical testing in the future. Fourth, it is anticipated that government subsidy could potentially mediate the CSR-CFP relationship, yet this hypothesis is rejected by the empirical investigation. As mentioned earlier, China's regulatory environment is unique, and there are other factors that may influence the mediating role of government subsidy. Considering the choice of research context, it must be acknowledged that the influence of the Chinese government on the CSR-CFP relationship can not be fully generalised and applied to other countries. Therefore, the role of government within the CSR-CFP relationship requires further investigation under different research contexts. Last but not least, although this study has adopted several variables to gauge the enterprise's CSR performance, they are considered as a whole. In future studies, the contribution of each stakeholder can be examined in greater detail.

## 6. Conclusions

By shifting the research perspective from developed countries to emerging economies, this study extends the applicability of CSR. In addition, through quantifying the impact of various stakeholders on CFP, this research partially mitigates the subjectivity inherent in previous methodologies, such as questionnaire-based surveys or reliance on third-party ratings. This study, furthermore, establishes a precedent for examining the potential mediating role of government in the CSR-CFP relationship. Despite the lack of statistical significance in the empirical results, the findings remain informative for future study. Last but not least, this study innovatively employs the threshold regression method to preliminarily prove the non-linear relationship between CSR and CFP. This challenges the traditional linear hypothesis connecting CSR and CFP, thereby paving the way for further refinement of associated theories and practices in the future.

Until now, numerous studies have made efforts to explore the relationship between enterprises' CSR implementation and their corresponding financial performance and have drawn various conclusions [9,21,57,58,94]. Considering the changeable and turbulent business environment, it is necessary to further clarify the relationship between CSR and CFP in different contexts, which will help contemporary enterprises grasp the balance between responsibility fulfilment and profitability [14]. This study builds on the stakeholder theory and transmits the research perspective from developing countries to emerging economies by using the data from Chinese Listed Companies from 2017 to 2022. The author

uses the panel data and adopts several approaches, including fixed effects regression and threshold regression to make causal inferences regarding the CSR-CFP relationship. Based on the above, the following conclusions are outputted by this research.

First, there is a positive relationship between enterprises' CSR initiatives and their financial performance, which coincides with the findings of prior studies [13,22,23]. Second, the government subsidy fails to function as a mediator within the CSR-CFP relationship. Specifically, both the conduction route of CSR-government subsidy and government subsidy–financial performance are rejected by this research. Third, although the overall relationship between CSR and CFP is positive, there is a non-linear relationship between them. As the level of CSR fulfilment increases, the CSR's contribution to CFP shows a flat-steep-flat trend. In sum, when a firm's CSR is at a medium level of fulfilment, its contribution to financial performance can be maximised.

**Author Contributions:** Conceptualization, X.L. and A.E.; methodology, X.L.; software, X.L.; validation, X.L., A.E. and Y.Z.; formal analysis, X.L.; investigation, X.L.; resources, A.E. and Y.Z.; data curation, X.L.; writing-original draft preparation, X.L.; writing-review and editing, A.E. and Y.Z.; visualization, X.L.; supervision, A.E.; project administration, Y.Z. All authors have read and agreed to the published version of the manuscript.

**Funding:** This research received no external funding.

**Institutional Review Board Statement:** Not applicable.

**Informed Consent Statement:** Not applicable.

**Data Availability Statement:** Publicly available datasets were analyzed in this study. The data can be found here: https://data.csmar.com.

**Conflicts of Interest:** The authors declare no conflict of interest.

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
