# Peer review of "The Impact of Corporate Social Responsibility Implementation on Enterprises’ Financial Performance—Evidence from Chinese Listed Companies"

_sustainability, doi:10.3390/su16051848_

Round 1

Reviewer 1 Report

Comments and Suggestions for Authors Overall, I think the study is up to the standard of the journal, but there are a few things to point out. (1) CSR-CFP research has been repeated for decades, and this study may be unique in that it uses an emerging market (China), finds non-linearity, and does not find an effect from the role of government. However, (2) I suspect that the variables used in the study are unique and different from many previous studies. I don't believe that one particular methodology is the correct answer, so I only raise the issue of comparability, i.e., if the results differ from previous studies, it could be due to only the variable differences. In general, using unique variables that have not been used in many previous studies should be noted as a limitation in terms of comparability. Comparability issues mean that there are problems in comparing the results of this study with those of existing studies. (3) This study is based on a Chinese sample. I suspect that due to the uniqueness of Chinese society, the impact of government in a Chinese sample can be different from the impact of government in a sample of other countries. This raises the issue of generalizability in the proof of your hypothesis. This can also be pointed out as a limitation. (4) There does not seem to be a logical reason for using a non-linear model, and vague statements such as "the condition of the current business environment" as in line 65 or the mixed results of previous studies as in section 2.2.3. seem insufficient. Based on my empirical test experiences, I intuitively agree that mixed results may mean that the relationship is not necessarily linear, as the results are sensitive to differences in study conditions, context, etc. I think this could be expressed in a more sophisticated way. (5) Is there a fundamental difference between a moderating effect and a mediating effect, i.e., is there a fundamental reason to choose one and not the other, e.g., that one allows you to look more closely at the structure of the variables than the other? You cite this as one of the contributions of this study, but the problem is that the lack of significant results may diminish the contribution of adopting a different methodology, because you are not able to look at any structure. Consider this a bit more, and if you agree it is a problem, include it in your discussion of limitations. (6) There are several places where the Error! Statements appear. These need to be corrected. Also, independent and dependent in Table 1 seem to be misplaced. (7) One shortcoming of this study is that, although a variety of CSR variables were employed, they only served to create a single interpretation of CSR. I suggest that each CSR variable be developed as a separate hypothesis in your future papers. (8) It would be more beneficial for the reader if the results in Table 9 could also be visualized in a graph, but this is just a suggestion. Comments on the Quality of English Language No specific comments.

Author Response

Comment #1: Overall, I think the study is up to the standard of the journal, but there are a few things to point out. (1) CSR-CFP research has been repeated for decades, and this study may be unique in that it uses an emerging market (China), finds non-linearity, and does not find an effect from the role of government.

Response #1:

  Thanks for your comment. This study endeavours to probe the CSR-CFP relationship in a relatively novel context, yielding distinct findings concerning the trajectory of this relationship and the influence of government intervention.  

Comment #2: However, (2) I suspect that the variables used in the study are unique and different from many previous studies. I don't believe that one particular methodology is the correct answer, so I only raise the issue of comparability, i.e., if the results differ from previous studies, it could be due to only the variable differences. In general, using unique variables that have not been used in many previous studies should be noted as a limitation in terms of comparability. Comparability issues mean that there are problems in comparing the results of this study with those of existing studies.

Response #2:

  We acknowledge that the variables to measure the CSR-CFP relationship are relatively unique and is different from some existing studies. Nevertheless, the choice of variables is based on the stakeholder theory and we believe that using those variables can overcome the inconsistency and subjectivity of some previous studies. Moreover, the numerical data from the official database can potentially increase the robustness of the outcome. However, as noted in the feedback comments, this may cause the issue of comparability. We have pointed out this concern in the limitation section (see line 666-671).

Comments #3: This study is based on a Chinese sample. I suspect that due to the uniqueness of Chinese society, the impact of government in a Chinese sample can be different from the impact of government in a sample of other countries. This raises the issue of generalizability in the proof of your hypothesis. This can also be pointed out as a limitation.

Response #3:

  Considering the special regulatory environment of China, the function of government within CSR-CFP relationship may not capture the full panorama, and it calls for further exploration in different regions. The problem of generalisability has been noted in the limitation section (see line 671-678).

Comment #4: There does not seem to be a logical reason for using a non-linear model, and vague statements such as "the condition of the current business environment" as in line 65 or the mixed results of previous studies as in section 2.2.3. seem insufficient. Based on my empirical test experiences, I intuitively agree that mixed results may mean that the relationship is not necessarily linear, as the results are sensitive to differences in study conditions, context, etc. I think this could be expressed in a more sophisticated way.

Response #4:

  In this study, a review of relevant literature reveals that prior studies mainly focused on the linear relationship between CSR and CFP, with some studies suggesting a neutral relationship between them. Considering the limited resources of enterprises, we hypothesise that a firm’s continuous fulfilment of CSR may not always have a straightforward positive or negative impact on its financial performance, but instead may exhibit a non-linear relationship. Existing theories on the CSR-CFP relationship only consider the linear relationship between the two and do not explore the potential non-linear dynamics, thereby creating a gap in theoretical understanding. Consequently, we include this potential theoretical limitation as part of the theoretical contribution of this study. We anticipate that relevant theoretical evidence will emerge through further research on the CSR-CFP relationship in the future. Also, we give further explanation of this in our manuscript (see line 267-276).

Comment #5: Is there a fundamental difference between a moderating effect and a mediating effect, i.e., is there a fundamental reason to choose one and not the other, e.g., that one allows you to look more closely at the structure of the variables than the other? You cite this as one of the contributions of this study, but the problem is that the lack of significant results may diminish the contribution of adopting a different methodology, because you are not able to look at any structure. Consider this a bit more, and if you agree it is a problem, include it in your discussion of limitations.

Response #5: 

  Thanks for your comments. Previous studies have basically focused on the government’s moderating role in the CSR-CFP relationship. Therefore, we aim to investigate whether the government can serve as a mediator in this relationship. In secton 2.2.1, we present potential pathways through which CSR influences CFP, specifically examining the role of government subsidy. Although the theoretical analysis suggests that government subsidy should mediate this relationship, the empirical results indicate insignificance, which could be attributed to the selection of variables and/or the research context. We believe that there is no inherent superiority between moderating and mediating effects, rather, differences lie in research perspectives. However, as suggested in the comments, we acknowledged in the limitation section that the role of government could be further explored particularly with respect to its potential mediating effect (see line 674-678).

Comment #6: There are several places where the Error! Statements appear. These need to be corrected. Also, independent and dependent in Table 1 seem to be misplaced.

Response #6:

  Thanks for identifying these errors. They have all been edited and corrected (see line 379, 407, Table1, 446, 459, 477, 500, 517, 518, 533, 544, 547, 556).

Comment #7: One shortcoming of this study is that, although a variety of CSR variables were employed, they only served to create a single interpretation of CSR. I suggest that each CSR variable be developed as a separate hypothesis in your future papers.

Response #7:

  Thanks for your comments. Based on your recommendation, we have addressed this issue in the limitation section (see line 679-681).

Comment #8: It would be more beneficial for the reader if the results in Table 9 could also be visualized in a graph, but this is just a suggestion.

Response #8:

  Thanks for your suggestion – we will of course endeavour to visualize the results in the future.

Reviewer 2 Report

Comments and Suggestions for Authors

Very positive assessment of the article, details in the report.

Author Response

Comments: In my opinion, the article can be published in its current form. I assess all aspects, i.e. literature review, research methodology, results and their usefulness and originality, positively. In particular, I want to highlight the following aspects:

  1. Very well defined and filled by the research gaps: practical and theoretical
  2. Precise theoretical part, a thorough review of the definitions and concepts in the field of CSR
  3. Very well developed point 2.2, with elaboration on the nature of the relationship between CSR and CFP
  4. The results obtained are original because they measure CSR from the perspective of various stakeholder groups: shareholders, employees, creditors, customers, suppliers, communities, government. Moreover, the study expands the application of CSR by shifting the research perspective from developed countries to developing countries.

Response:

  We’d like to express our gratitude for your review and positive assessment of our article. We sincerely appreciate your time and effort in evaluating all aspects of our work. Your encouraging feedback strengthens our confidence to produce more high-quality research that contribute meaningfully to current business and managerial environment. Thanks again for your valuable assessment and feedback to our work.

Reviewer 3 Report

Comments and Suggestions for Authors

The structure of the article is well constructed. The work contains both theoretical elements and an empirical part.

The cited publications are carefully selected. The selected publications were selected appropriately. The authors selected articles that are not only related to CSR issues, but also specifically refer to the research conducted. The authors mostly refer to new publications.

The selected statistical methods are well described. As for the choice of individual methods, there can also be no objections. The results are presented clearly.

The conclusions are clearly formulated. The publication includes a discussion relating to other studies. The article also indicates the limitations of the study. Individual elements create a coherent whole and increase the value of the publication.

The article is written correctly in English.

The article may be published as is. However, it is worth looking at some sentences from a stylistic perspective and maybe they could be simplified a bit. This is only a suggestion.

Comments on the Quality of English Language

The article is written correctly in English.

The article may be published as is. However, it is worth looking at some sentences from a stylistic perspective and maybe they could be simplified a bit. This is only a suggestion.

Author Response

Comment: 

The structure of the article is well constructed. The work contains both theoretical elements and an empirical part.

The cited publications are carefully selected. The selected publications were selected appropriately. The authors selected articles that are not only related to CSR issues, but also specifically refer to the research conducted. The authors mostly refer to new publications.

The selected statistical methods are well described. As for the choice of individual methods, there can also be no objections. The results are presented clearly.

The conclusions are clearly formulated. The publication includes a discussion relating to other studies. The article also indicates the limitations of the study. Individual elements create a coherent whole and increase the value of the publication.

The article is written correctly in English.

The article may be published as is. However, it is worth looking at some sentences from a stylistic perspective and maybe they could be simplified a bit. This is only a suggestion

Response:

   We appreciate your suggestion regarding the stylistic aspects of certain sentences to which we carefully reviewed them with the aim of simplification while maintaining clarity and coherence. We have made several adjustments to make our manuscript more rigorous and complete. Once again, we sincerely thank you for your constructive comments and feedback to our manuscript.